# ATP8B1 Deficiency Results in Elevated Mitochondrial Phosphatidylethanolamine Levels and Increased Mitochondrial Oxidative Phosphorylation in Human Hepatoma Cells

**DOI:** 10.3390/ijms232012344

**Published:** 2022-10-15

**Authors:** Valentina E. Gómez-Mellado, Jung-Chin Chang, Kam S. Ho-Mok, Carmen Bernardino Morcillo, Remco H. J. Kersten, Ronald P. J. Oude Elferink, Arthur J. Verhoeven, Coen C. Paulusma

**Affiliations:** 1Amsterdam UMC, University of Amsterdam, Tytgat Institute for Liver and Intestinal Research, Meibergdreef 69, 1105 BK Amsterdam, The Netherlands; 2Amsterdam Gastroenterology Endocrinology Metabolism, 1105 AZ Amsterdam, The Netherlands; 3Department of Biomolecular Health Sciences, Faculty of Veterinary Medicine, Utrecht University, 3584 CS Utrecht, The Netherlands

**Keywords:** phosphatidylethanolamine, flippase, ATP8B1, mitochondria, OXPHOS, LDLR, PFIC

## Abstract

ATP8B1 is a phospholipid flippase that is deficient in patients with progressive familial intrahepatic cholestasis type 1 (PFIC1). PFIC1 patients suffer from severe liver disease but also present with dyslipidemia, including low plasma cholesterol, of yet unknown etiology. Here we show that ATP8B1 knockdown in HepG2 cells leads to a strong increase in the mitochondrial oxidative phosphorylation (OXPHOS) without a change in glycolysis. The enhanced OXPHOS coincides with elevated low-density lipoprotein receptor protein and increased mitochondrial fragmentation and phosphatidylethanolamine levels. Furthermore, expression of phosphatidylethanolamine N-methyltransferase, an enzyme that catalyzes the conversion of mitochondrial-derived phosphatidylethanolamine to phosphatidylcholine, was reduced in ATP8B1 knockdown cells. We conclude that ATP8B1 deficiency results in elevated mitochondrial PE levels that stimulate mitochondrial OXPHOS. The increased OXPHOS leads to elevated LDLR levels, which provides a possible explanation for the reduced plasma cholesterol levels in PFIC1 disease.

## 1. Introduction

ATP8B1 belongs to the P4-ATPase family of proteins that catalyze the transport of phospholipids from the exoplasmic- to the cytosolic leaflet of biological membranes, an activity essential for the creation of membrane asymmetry [1]. The establishment and maintenance of an asymmetric distribution of phospholipids is crucial for membrane protein activity and barrier function as well as for the biogenesis, fission and fusion of membrane vesicles [2,3]. Mutations in the *ATP8B1* gene cause liver disease characterized by a continuous disease spectrum from intermittent Benign Recurrent Intrahepatic Cholestasis type 1 (BRIC1) to severe progressive Familial Intrahepatic Cholestasis type 1 (PFIC1) [4]. PFIC1 is a progressive autosomal recessive disorder of early onset, which is characterized by impaired bile formation (i.e., cholestasis) that advances to severe, end-stage liver disease, requiring liver transplantation before the second decade of life [5,6]. PFIC1 is, however, not restricted to the liver, as evidenced by the wide range of extrahepatic manifestations in patients, including diarrhea, pulmonary problems and hearing loss [7,8,9]. We and others have previously studied the molecular mechanisms that are likely to contribute to these extrahepatic phenotypes, which highlighted a role for ATP8B1 in the targeting of multiple membrane proteins, including apical ectoenzymes (CD13, CD26) [10], the intestinal apical bile acid uptake transporter (SLC10A2) and CFTR in intestinal and pulmonary epithelial cells [11,12], all of which were reduced when ATP8B1 expression was reduced.

In addition to the abovementioned phenotypes, and in contrast to other forms of cholestasis, low/normal γ-glutamyl transpeptidase PFIC1 patients present with dyslipidemia, characterized by hypertriglyceridemia, low high-density lipoprotein and low plasma cholesterol levels, of which the etiology is not understood [13,14,15]. Low-density lipoprotein (LDL) levels were normal, however, the LDL particles were small and dense, low in cholesterol ester content and highly enriched in triglycerides [13], rendering low affinity for the LDL receptor (LDLR) [16]. We here studied the hypothesis that ATP8B1 is involved in the regulation of plasma membrane localization of the LDLR, which plays an essential role in the maintenance of plasma cholesterol levels (reviewed in [17]). We show that ATP8B1 knockdown coincides with increased LDLR protein levels that, unexpectedly, is associated with an increased preference for mitochondrial oxidative phosphorylation.

## 2. Results 

### 2.1. ATP8B1 Knockdown in HepG2 Cells Leads to Elevated Levels of LDLR 

Low-density lipoprotein receptor (LDLR) levels were analyzed in HepG2 cells, in which *ATP8B1* expression was reduced by ~75% using lentiviral knockdown (Figure 1A). Western analysis showed that ATP8B1 protein levels were reduced by ~65% (Figure 1B,C). Unexpectedly, ATP8B1 knockdown coincided with a ~50% increase in LDLR protein levels (Figure 1B,C). Recently, Khan et al. [18] showed the upregulation of LDLR in HepG2 cells that were treated with dichloroacetic acid (DCA) [19], an inhibitor of pyruvate dehydrogenase kinase-1 (PDK1). This compound has been clinically used to reduce plasma LDL-cholesterol levels in patients with familial hypercholesterolemia, diabetes and hyperlipoproteinemia [20,21]. Inhibition of PDK1 relieves the block on the conversion of pyruvate to acetyl-CoA, thus switching glucose metabolism from aerobic glycolysis to mitochondrial oxidative phosphorylation (OXPHOS). To investigate whether such a mechanism could be involved in the elevation of LDLR in ATP8B1 knockdown cells, we analyzed LDLR protein levels after DCA application. Indeed, as previously shown [18], DCA induced higher LDLR protein levels in control cells (Figure 1D,E).

In ATP8B1 knockdown cells, however, the elevated LDLR protein levels did not (further) increase upon DCA treatment (Figure 1D,E). The similarity between the effects of ATP8B1 knockdown and DCA suggested shared common mediators or mechanisms to enhance LDLR expression, possibly induced by a metabolic switch from glycolysis to mitochondrial OXPHOS.

### 2.2. ATP8B1 Knockdown HepG2 Cells Show Increased Mitochondrial OXPHOS

To address the possibility of a metabolic switch in ATP8B1 knockdown cells, we studied mitochondrial respiration by Seahorse extracellular flux analysis. When cells were analyzed under glucose-supplemented conditions, ATP8B1 knockdown cells indeed showed a much higher oxygen consumption rate (OCR) compared to control cells (Figure 2A). Inhibition of mitochondrial respiration by oligomycin, which blocks the mitochondrial ATP synthase, revealed a >2-fold increase in ATP-coupled respiration in the knockdown cells compared to control cells (Figure 2B).

The leak-associated OCR, a measure of a.o. H^+^ leak over the mitochondrial inner membrane [22], was 1.4-fold increased, while the uncoupled OCR, i.e., the difference between maximal and basal respiration (exposed after addition of the proton ionophore FCCP), was ~4-fold elevated in the knockdown cells (Figure 2B). The elevated basal OCR in ATP8B1 knockdown cells was not accompanied by a reduction in the basal extracellular acidification rate (ECAR) (Figure 2C and Appendix A), suggesting no reciprocal reduction in glycolysis. Since the ECAR is the result of both glycolysis and TCA cycle activity, we also determined the oligomycin-induced increase in ECAR (Appendix A) and measured the lactate concentration in the medium of ATP8B1 knockdown cells (Appendix A). These results supported an increased OXPHOS without a change in glycolysis in ATP8B1 knockdown cells (Appendix A), suggesting that total ATP consumption was higher in ATP8B1-deficient cells. From the extracellular flux data, we also estimated the ATP turnover rates (*Ј*_ATP_) for OXPHOS and glycolysis [23,24,25]. These were ~20% and ~80%, respectively, of total ATP production in control cells and shifted to ~40 and ~60% in ATP8B1 knockdown cells (Figure 2D). Altogether these data indicate that in ATP8B1 knockdown HepG2 cells the contribution of mitochondrial OXPHOS was increased.

### 2.3. ATP8B1 Knockdown Coincides with a Shift towards Mitochondrial ß-Oxidation

Next, we investigated whether the increased OXPHOS correlated with a change in substrate utilization. When glucose was replaced by the short-chain fatty acid (and mitochondrial substrate) octanoate, a similar though less pronounced elevation of oxygen consumption was observed in the knockdown cells (Appendix A) indicating normal or enhanced capacity to oxidize fatty acids. Next, we studied mitochondrial respiration under no-substrate conditions with and without etomoxir-mediated inhibition of carnitine palmitoyltransferase 1A (CPT1A). CPT1A catalyzes the first step in the entry of long- and medium-chain fatty acids into the mitochondrial matrix for mitochondrial ß-oxidation [26]. When no substrate was added, oxygen consumption was again strongly increased in the knockdown cells, in particular under uncoupled conditions (Figure 3A,B).

Importantly, etomoxir almost completely nullified the difference between control and ATP8B1 knockdown cells (Figure 3C–E), suggesting that the increased OXPHOS in ATP8B1 knockdown cells was the result of enhanced mitochondrial ß-oxidation of endogenous fatty acids.

### 2.4. ATP8B1 Knockdown Cells Have Increased Mitochondrial Fragmentation and Elevated Mitochondrial PE Levels

To obtain a mechanistic explanation for increased OXPHOS utilization in ATP8B1 knockdown cells, we studied mitochondrial mass, fragmentation and lipid composition. Genomic PCR analysis of the mtDNA coding for NADH-ubiquinone oxidoreductase chain 1 (complex 1 subunit) as well as Western analysis of electron transport chain complex I-V protein markers showed no significant differences between control and ATP8B1 knockdown cells (Figure 4A–C), indicating no difference in mitochondrial mass. Immunofluorescent detection of the mitochondrial outer membrane-localized protein TOM20 suggested that ATP8B1 knockdown cells had a more fragmented mitochondrial network (Figure 4D). Indeed, quantification of the mitochondrial area demonstrated that the mitochondria from the knockdown cells were smaller, even after correction for cell swelling (Figure 4E,F).

Importantly, analysis of the mitochondrial phospholipid composition revealed that the relative abundance of cardiolipin (CL), phosphatidylinositol (PI), phosphatidylcholine (PC) and sphingomyelin (SM) were unaffected, while phosphatidylethanolamine (PE) levels were increased from 9.8% in control cells to 13.7% in the knockdown cells (Figure 5A). In addition, mitochondrial cholesterol content was also unaffected (Appendix A). PE synthesized in the mitochondria is converted to PC by the enzyme phosphatidylethanolamine N-methyltransferase (PEMT) [27], and PEMT deficiency has been shown to be associated with increased mitochondrial fragmentation, PE levels and oxidative phosphorylation [28]. Intriguingly, ATP8B1 knockdown HepG2 cells displayed an ~50% reduction in PEMT levels (Figure 5B,C), a phenotype that was also observed when the cells were immunostained for PEMT (Figure 5D).

The reduced PEMT levels did not coincide with reduced *PEMT* mRNA expression, indicating that this involves a posttranslational event (Figure 5E). Staining of PEMT in HepG2 cells that ectopically expressed ATP8B1-eGFP showed abundant plasma membrane staining of ATP8B1eGFP, but also punctate staining that, although in close proximity of PEMT-positive puncta, never co-stained with PEMT (Appendix A). Altogether these data show that ATP8B1 knockdown cells have increased mitochondrial fragmentation, increased mitochondrial PE levels and reduced PEMT levels.

## 3. Discussion

In contrast to normal cells, most cancer cells, including HepG2 cells, meet their energy demands (despite sufficient oxygen pressure) via cytosolic aerobic glycolysis in which glucose is converted to lactate and ATP. This well-established Warburg effect is inefficient, as it consumes large amounts of glucose to produce relatively low amounts of ATP (one glucose molecule yields two molecules of ATP), as opposed to the mitochondrial tricarboxylic acid (TCA) cycle and oxidative phosphorylation in which one molecule of glucose yields 38 molecules of ATP [29]. Intriguingly, we show here that knockdown of the phospholipid flippase ATP8B1 in HepG2 cells weakens the Warburg phenotype and coincides with a metabolic reprogramming in which the mitochondrial OXPHOS pathway is strongly increased without a change in glycolysis, mitochondrial mass or electron transport chain complex proteins.

The increased OXPHOS coincided with elevated mitochondrial phosphatidylethanolamine (PE) levels, whereas levels of other mitochondrial lipids, including cardiolipin, were unaffected. PE, such as cardiolipin, is a non-bilayer, fusogenic lipid and enriched in the mitochondrial inner membrane, where it contributes to the structure of the cristae in which the protein complexes of the electron transport chain (ETC) reside (reviewed in [30,31]). Cellular PE is produced via two pathways, i.e., the cytidine 5′-diphosphate-ethanolamine (Kennedy) pathway in the endoplasmic reticulum (ER) [32], and in mitochondria from phosphatidylserine (PS) in a reaction catalyzed by PS decarboxylase 1 (PSD1) (reviewed in [33]). In addition to residence in the cristae, mitochondrial PE is exchanged at the mitochondrial-ER contact sites, i.e., the mitochondrial-associated membrane (MAM) [34], where PE is methylated to phosphatidylcholine by the enzyme phosphatidylethanolamine N-methyltransferase (PEMT) [27]. Van der Veen et al. [28] demonstrated that PEMT deficiency associated with increased mitochondrial fragmentation and elevated (~20%) mitochondrial PE levels increased ETC protein complex activity and increased OXPHOS in mouse hepatocytes and hepatoma cells. In addition, they showed a positive correlation between mitochondrial PE and ATP content and a negative correlation between PEMT activity and ATP content [28]. More recently, Heden et al. [35] showed that PSD1 overexpression in C2C12 cells or in mouse skeletal muscle resulted in elevated mitochondrial PE levels and a concomitant increased oxygen consumption. Reversibly, skeletal muscle-specific knockout of PSD1 or knockdown in C2C12 cells resulted in reduced mitochondrial PE levels, reduced ETC protein complex activities and reduced oxygen consumption rates [35]. These observations underscore the importance of mitochondrial PE content for the activity of the ETC and mitochondrial respiration. Importantly, ATP8B1 knockdown cells phenocopy PEMT deficient cells, in that they also show mitochondrial fragmentation, increased mitochondrial PE, and increased OXPHOS activity. In line with this, ATP8B1 knockdown cells show a ~50% reduction in PEMT protein levels with a concomitant ~28% increase in mitochondrial PE levels. The elevated mitochondrial PE levels might also be responsible for the increased proton leak observed in ATP8B1 knockdown cells (Figure 2B), although this can also be the result of the much higher activity of the ETC chain. Altogether these data provide a likely explanation for the increased OXPHOS activity in ATP8B1 knockdown cells.

The shift towards mitochondrial OXPHOS in ATP8B1 knockdown cells coincided with an increased dependency on mitochondrial ß-oxidation. Our Seahorse analyses showed that when no-substrate was provided, and the cells thus depended on substrate intake and/or release from cellular stores, respiration rates were strongly increased in the knockdown cells. This increase was completely inhibited by etomoxir, indicating that the cells had a preference for mitochondrial ß-oxidation. A possible explanation is that the cells respond to PE-induced enhancement of the ETC activity by increasing mitochondrial ß-oxidation. In line with this, we observed that *CD36* expression was induced in ATP8B1 knockdown cells (Appendix A), which is involved in the uptake of medium- and long-chain fatty acids (reviewed in [36]), and which serve as substrates for the ß-oxidation.

At the moment, it is unclear why ATP8B1 knockdown cells have reduced PEMT levels. One explanation could be that ATP8B1 has a role in the targeting of PEMT to the mitochondrial-associated membrane. Alternatively, as a phospholipid flippase for i.e., PE, PS and PC [37,38,39], ATP8B1 deficiency could interfere with membrane fluidity, including that of the endoplasmic reticulum (ER), which could result in destabilization and premature degradation of PEMT. Destabilization and premature degradation of several ER resident proteins was recently shown in *S. cerevisiae*, in which PC synthesis was impaired due to inactivation of the yeast PEMT homologue Opi3 [40]. The authors showed that reduced PC synthesis (from PE) in *opi3*Δ cells resulted in reduced membrane fluidity in the ER, which is associated with a destabilization of ER proteins, providing a likely explanation for the premature degradation of selected ER residents. In line with these findings is the observation that a decreased PC/PE ratio contributes to reduced fluidity (due to phase separation of PE) of the membrane [41]. Thus, ATP8B1 deficiency could result in affected ER membrane fluidity and consequent premature degradation of PEMT, resulting in an increased mitochondrial PE content.

In conclusion, we show that ATP8B1 deficiency is associated with an increased mitochondrial respiratory capacity. From our data, we hypothesize that ATP8B1 deficiency leads to reduced PEMT levels, which results in an elevation of mitochondrial PE content (Figure 6). Increased mitochondrial PE levels stimulate the ETC activity which results in an increased OXPHOS utilization. It is likely that the elevated LDLR levels in ATP8B1 knockdown cells are the consequence of this metabolic change, as DCA-induced glycolysis-to-OXPHOS switching has been shown to be associated with enhanced expression of LDLR (our study and [18]). Whether such a mechanism contributes to the reduced plasma cholesterol levels (and dyslipidemia in general) in PFIC1 patients awaits further studies, preferably in PFIC1 patients.

## 4. Materials and Methods

### 4.1. Cell Culture

The human hepatocellular carcinoma cell line HepG2 (ATCC, HB-8065) was cultured in high (4.5 g/L) glucose Dulbecco’s modified Eagle’s medium (DMEM) (Lonza, Geleen, The Netherlands) supplemented with 10% Fetal Bovine Serum (FBS) (Lonza), 2 mM L-glutamine (Lonza), 100 U/mL penicillin (Lonza), and 100 U/mL streptomycin (Lonza) at 37 °C in a 10% CO_2_ humidified atmosphere. The knockdown cell line for ATP8B1 was established by lentiviral transduction with the validated short-hairpin RNA (shRNA) vector to *ATP8B1* (TRCN0000050127) from the Mission shRNA Library (Sigma-Aldrich, Saint Louis, MO, USA). ATP8B1 knockdown cells did not show any growth defects as witnessed by trypan blue staining of the cultures. The non-targeting hairpin SHC002 in pLKO.1-puro (CAACAAGATGAAGAGCACCAA) was included as a control. Lentiviral construct to enhanced green fluorescent protein (eGFP)-tagged *ATP8B1* (*ATP8B1-eGFP*) was described previously [38].

### 4.2. Quantitative RT-PCR

Total RNA was isolated from cells using TriPURE reagent (Invitrogen, Waltham, MA, USA). cDNA was synthesized from 1–2 µg total RNA with random hexamers, oligo-dT_12–18_ primer and Superscript III RT (Invitrogen). Real-time PCR measurements were performed on a Lightcycler 480 (Roche, Basel, Switserland) with Fast Start DNA MasterPlus SYBR Green I kit (Roche). Expression levels in HepG2 cells were calculated with the LinRegPCR software [42] and were normalized to the geometric means of three reference genes (RPLP0, cyclophilin, HPRT). Genomic DNA was isolated from HepG2 cells by incubation in lysis buffer (200 mM sodium chloride, 0.2% sodium dodecyl sulphate (SDS), 5 mM EDTA, 100 µg/mL proteinase K (Merck, Amsterdam, The Netherlands), 100 mM Tris, pH = 8.0) for 3 h at 55 °C. DNA was precipitated with isopropanol, washed with 70% ethanol and dissolved in TE. Mitochondrial DNA content was quantified by real-time PCR and normalized to HepG2 single-copy gene *ADCY10*. Oligo sequences are in Appendix A.

### 4.3. SDS Polyacrylamide Gel Electrophoresis (SDS-PAGE) and Western Blotting

Cells were lysed in radioimmunoprecipitation assay (RIPA) buffer (150 mM sodium chloride, 1% Triton X-100, 0.5% sodium deoxycholate, 0.1% sodium dodecyl sulphate (SDS), 50 mM Tris, pH = 8.0) supplemented with EDTA-free protease inhibitor cocktail (Roche) and PhosStop phosphatase inhibitor cocktail (Roche). Protein concentrations were determined by Pierce Bicinchoninic Acid (BCA) Protein Assay. Extracts were fractionated by SDS-PAGE and transferred to Immobilon-P Polyvinylidene difluoride (PVDF) membranes (Millipore, Burlington, MA, USA) by a semi-dry blotting system (Bio-Rad) using either 10 mM N-cyclohexyl-3-aminopropanesulfonic acid (CAPS), pH 10.5/10–15% methanol buffer or 25 mM ethanolamine-glycine, pH 11 buffer. Membranes were blocked for 1 h at room temperature (RT) in blocking solution (PBS/0.05% Tween-20 containing 5% low-fat milk (Nutricia Profitarplus, Zoetermeer, The Netherlands)), followed by incubation with antibodies. Primary antibodies used: rabbit anti-LDLR (BioVision, Milpitas, CA, USA), anti-OXPHOS complex (Abcam, Cambridge, UK), rabbit anti-GAPDH (Millipore, ABS16), mouse anti-actin (Sigma), rabbit anti-ATP8B1 [43], rabbit anti-ATP1A1 [44] and rabbit ant-PEMT (SABbiotech, Greenbelt, MD, USA). Immune complexes were visualized with peroxidase-conjugated goat-anti-rabbit or mouse IgGs (Bio-Rad, Lunteren, The Netherlands), developed with homemade enhanced chemiluminescence reagent (100 mM Tris–HCl, pH 8.5, 1.25 mM luminol, 0.2 mM p-coumarin and freshly added 3 mM H_2_O_2_). Images were acquired with ImageQuantTM LAS 4000. Densitometric analysis was performed with ImageJ.

### 4.4. Indirect Immunofluorescence and Mitochondrial Network Analysis

HepG2 cells were grown on glass coverslips and fixed in paraformaldehyde 3.7% during 15 min at RT. Fixed cells were permeabilized with PBS/0.1% triton X-100 and incubated with rabbit anti-LDLR (BioVision), rabbit anti-TOM20 (Santa Cruz; FL-145) or rabbit ant-PEMT (SABbiotech) for one hour at RT. Cells were extensively washed in PBS/0.1% Triton X-100 amidst antibody incubations. Immuno-reactivity was visualized with goat anti-rabbit Alexa 594 (Molecular Probes). Sections were mounted in Vectashield/DAPI (Vector Laboratories, Newark, CA, USA) and images were acquired on a Leica TCS SP8 X laser scanning microscope with an HC APO 63x/1.4 oil CS2 immersion lens.

The mitochondrial network (visualized by TOM20 staining) was evaluated in three independent experiments using ImageJ and the Mito Morphology Macro developed for ImageJ by Dagda et al. [45] (http://imagejdocu.tudor.lu/doku.php?id=plugin:morphology:mitochondrial_morphology_macro_plug-in:start#installation). In short, images were acquired of stained cells (3–4 cells per image) followed by the setting of thresholds, excluding very high (punctated) signals and background fluorescence. All images were scored blinded and in each experiment 3–5 images were analyzed. The average mitochondrial area/perimeter ratio was normalized to circularity to account for mitochondrial swelling as recommended by Dagda et al. [45], resulting in a mitochondrial connectivity index.

### 4.5. Extracellular Flux Analysis

Oxygen consumption rate (OCR) and extracellular acidification rate (ECAR) were analyzed by a Seahorse XF96 flux analyzer (Agilent, Santa Clara, CA, USA). Cells were grown in low (10 mM) glucose-containing medium. For Seahorse analysis, cells were washed in HBSS and then incubated in HBSS/0.1% BSA for 30 min at 37 °C (to deplete endogenous substrate), after which the cells were incubated in HBSS supplemented with 5.5 mM glucose/0.1% BSA, 125 µM octanoate/0.1% BSA, or no substrate (0.1% BSA) for 30 min at 37 °C. Subsequently, cells were equilibrated in the Seahorse analyzer for 20 min at 37 °C and baseline OCR was measured, after which 1.6 µM oligomycin A (port A), 1.1 µM FCCP (port B), and 4.62 μM antimycin A/2.31 μM rotenone (port C) were sequentially added (all chemicals from Sigma-Aldrich). Where indicated cells were incubated in the presence of 3 µM etomoxir (Cayman Chemical, Ann Arbor, MI, USA). ATP-linked, leak-associated OCR and uncoupled respiration rates were calculated as described previously [23]. Rates were corrected for protein content of each well. Estimation of ATP production rates were calculated as described previously [23,24,25].

### 4.6. Isolation of Mitochondria and High-Performance Thin-Layer Chromatography

Mitochondria were isolated as previously described (basic protocol 2 in [46]). Phospholipids were extracted from mitochondrial fractions by Bligh and Byer extraction [47]. Lipid films were dissolved in chloroform/methanol (1:2) and were run on silica gel 60 plates (Merck, Darmstadt, Germany) as previously described [48]. Briefly, silica plates were pre-run in methanol/ethylacetate (3:2) and dried for 10 min at 135 °C, after which lipid samples were spotted and run in a buffer containing chloroform/ethanol/milliQ/triethylamine (3:3.5:0.7:3.5). The following phospholipids were included as standards: cardiolipin (16:0, bovine heart, ICN), phosphatidyl(L-α)ethanolamine (porcine brain, Avanti), phosphatidyl(L-α)inositol (bovine liver, Sigma-Aldrich), phosphatidyl(L-α)choline (egg yolk, Sigma-Aldrich), sphingomyelin (porcine brain, Avanti Polar Lipids, Alabaster, AL, USA). The plate was dried for 3 min at 135 °C, after which the lipid spots were visualized by charring; for this, plates were incubated for 1 min in Cu(II)acetate/Cu(II) sulphate solution, after which the temperature of the plate was gradually increased from 60 to 160 °C. Spot densities were quantified using ImageJ software.

### 4.7. Data Presentation and Statistical Analysis

Graphs were generated, and statistical analyses were performed using GraphPad Prism version 9.1.0 software. Statistical tests used are indicated in the figure legends.

## Figures and Tables

**Figure 1 ijms-23-12344-f001:**
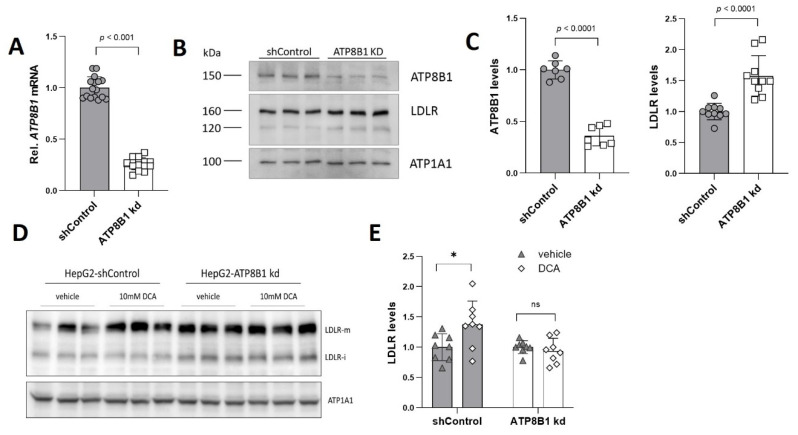
ATP8B1 knockdown coincides with increased LDLR expression. (**A**) *ATP8B1* mRNA expression in ATP8B1 knockdown HepG2 cells. Data were normalized to shControl cells and are expressed as means +/− s.e.m. of 16 replicates from 5 independent experiments. Statistical analysis by an unpaired *t*-test.; (**B**) LDLR protein levels are increased in ATP8B1 kd HepG2 cells. Western blot of total cell lysates from a representative experiment is shown. ATP1A1 is included as a transfer control; (**C**) Densitometric analysis of ATP8B1 and LDLR levels. Data are expressed as relative mean protein levels normalized to ATP1A1 ± standard deviation of 7–10 replicates from 2–3 independent experiments. Statistical analysis by an unpaired *t*-test; (**D**) Representative Western blot of total cell lysates showing the effect of DCA (10 mM, 48 h) on LDLR levels. LDLR-m, mature form of LDLR; LDLR-I, immature form of LDLR; (**E**) Quantification of LDLR protein levels after DCA challenge. Data were normalized to the vehicle control of each cell line and are expressed as means +/− s.d. of 8 replicates from 3 independent experiments. Statistical analysis by a multiple paired *t*-test; * *p* < 0.05; ns, not significant.

**Figure 2 ijms-23-12344-f002:**
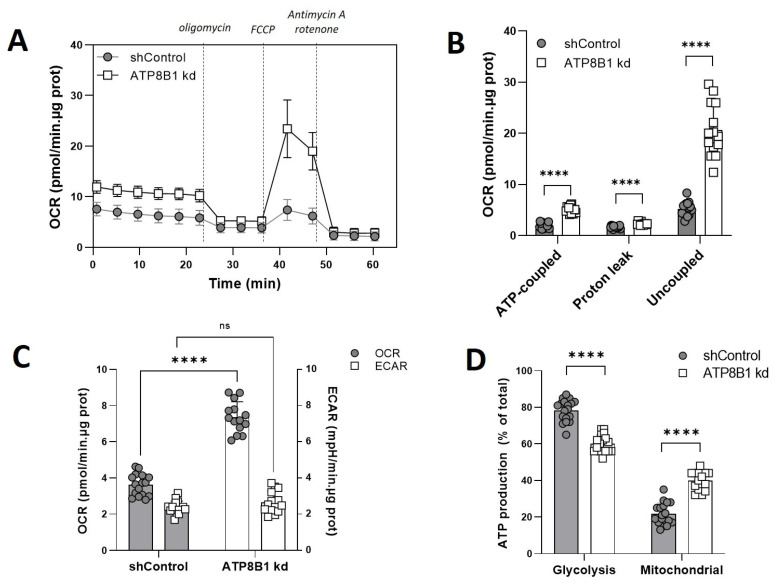
ATP8B1 knockdown coincides with increased mitochondrial OXPHOS. (**A**) Cells were incubated with 5.5 mM glucose and the oxygen consumption rate (OCR) was measured in a Seahorse analyzer. Dotted lines indicate the addition of different compounds to modulate mitochondrial respiration. Profiles shown are means +/− s.d. of 14–17 replicates of 2 independent experiments; (**B**) Quantification of ATP-coupled-, leak-associated and uncoupled OCR in the presence of glucose. Data are extracted from the data shown in (**A**), corrected for the non-mitochondrial OCR, and are expressed as means +/− s.d. of 14–17 replicates of 2 independent experiments; **** *p* < 0.0001 by multiple unpaired *t*-test; (**C**) Basal OCR compared to basal ECAR. Data are expressed as means +/− s.d. of 13–16 replicates from 2 independent experiments; **** *p* < 0.0001 by 2way ANOVA with Bonferroni’s correction for multiple testing; ns, not significant; (**D**) Estimated ATP production rate (in pmol/min.well) was calculated as described in M&M and is expressed as mean percentage of total ATP production +/− s.d. of 17–18 replicates from 2 independent experiments. **** *p* < 0.0001 by multiple unpaired *t*-test.

**Figure 3 ijms-23-12344-f003:**
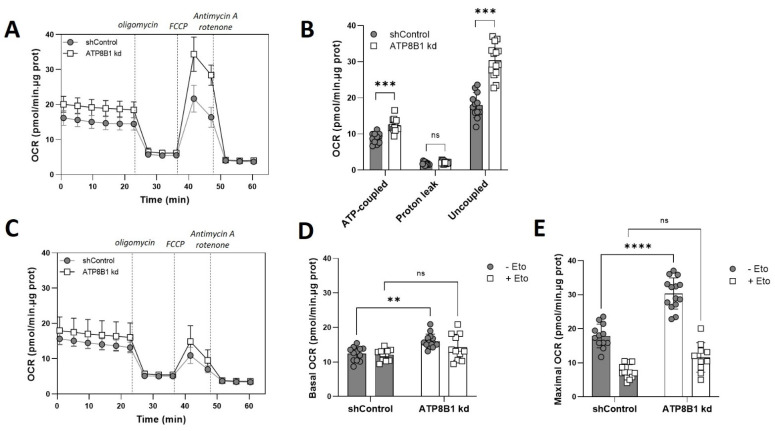
ATP8B1 knockdown coincides with an increased mitochondrial ß-oxidation. (**A**) Oxygen consumption rate (OCR) in the absence of added substrate. Profiles shown are means +/− s.d. of 12–14 replicates of 2 independent experiments; (**B**) Quantification of ATP-coupled-, leak-associated and uncoupled OCR when no substrate was added. Data are extracted from the data shown in (**A**), corrected for non-mitochondrial OCR, and are expressed as means +/− s.d.; *** *p* < 0.0001 by multiple unpaired *t*-test; (**C**) Oxygen consumption rate (OCR) in the absence of added substrate with 3 µM etomoxir. Profiles shown are means +/− s.d. of 12–14 replicates of 2 independent experiments; (**D**,**E**) Quantification of basal (**D**) and maximal (**E**) respiration (corrected for non-mitochondrial respiration) in the absence of added substrate with/without etomoxir (Eto). Data are expressed as means +/− s.d. of 11–14 replicates from 2 independent experiments. **** *p* < 0.0001, ** *p* < 0.005 by 2way ANOVA with Tukey’s correction for multiple testing; ns, not significant.

**Figure 4 ijms-23-12344-f004:**
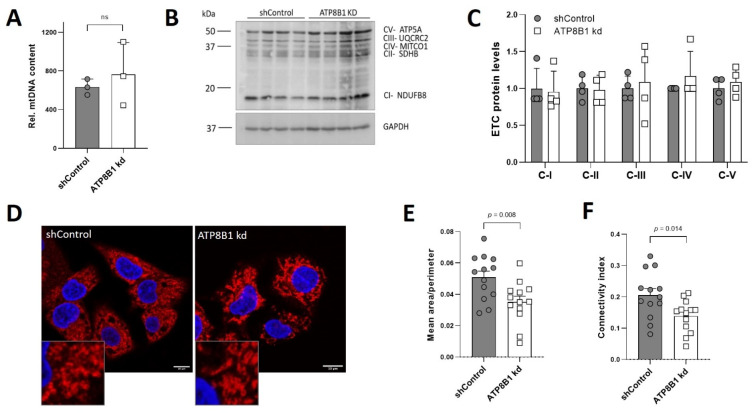
No change in mitochondrial mass and increased mitochondrial fragmentation in ATP8B1 knockdown cells. (**A**) Genomic LC-PCR analysis of the mtDNA *NADH-ubiquinone oxidoreductase chain 1.* Data are expressed as mean +/− s.d. of 3 independent DNA isolations. Statistical analysis by an unpaired *t*-test; ns, not significant; (**B**) Western blot analysis from total cell lysates of electron transport chain complex (ETC) I-V protein markers; (**C**) Densitometric analysis of ETC protein levels. Data were corrected for GAPDH, normalized to shControl cells and expressed as means +/− s.d. of 4 independent experiments. No statistical significance by multiple unpaired *t*-test; (**D**) Confocal analysis of TOM20 staining in red and nuclear dapi in blue. Bar = 10 nm; (**E**,**F**) Quantification of mitochondrial fragmentation. HepG2 cells were stained for TOM20 and the mitochondrial network was quantified as described in the materials and methods. Ratio of mean mitochondrial area over perimeter (**E**) and mitochondrial connectivity (**F**), representing average mitochondrial area/perimeter ratio normalized to circularity to account for mitochondrial swelling, are shown. Data are expressed mean values +/− s.e.m. of 13 images (3–4 cells/image) from 3 independent experiments. Statistical analysis by an unpaired *t*-test.

**Figure 5 ijms-23-12344-f005:**
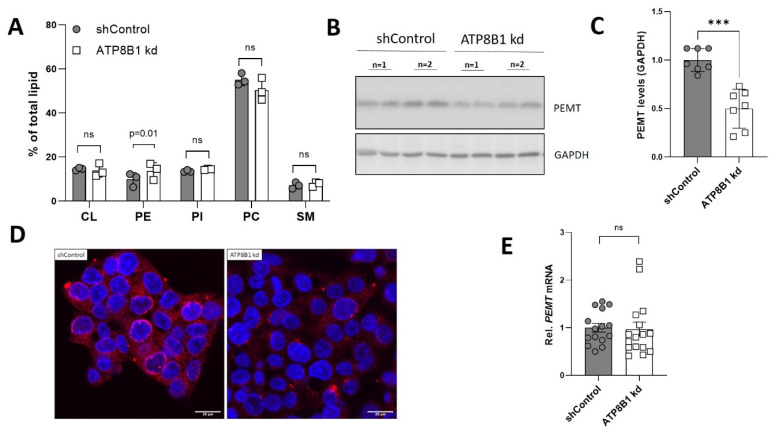
ATP8B1 knockdown cells have increased mitochondrial phosphatidylethanolamine content and reduced phosphatidylethanolamine N-methyltransferase (PEMT) levels. (**A**) Mitochondrial lipid species expressed as percentage of total mitochondrial lipid. Data are expressed as mean +/− s.d. of 3 independent mitochondria isolations. Statistical analysis by multiple unpaired *t*-test; (**B**) Representative Western blot showing PEMT protein levels in total cell lysates of control and ATP8B1 knockdown HepG2 cells; (**C**) Densitometric analysis of PEMT levels; data are expressed as mean +/− s.d. of 6 replicates from 3 independent experiments. Statistical analysis by an unpaired *t*-test. *** *p* = 0.0001; (**D**) Confocal analysis of PEMT staining in red and nuclear dapi in blue. Bar = 20 nm; (**E**) *PEMT* mRNA expression in ATP8B1 knockdown HepG2 cells. Data were normalized to shControl cells and are expressed as means +/− s.e.m. of 15 replicates from 5 independent experiments. Statistical analysis by an unpaired *t*-test.

**Figure 6 ijms-23-12344-f006:**
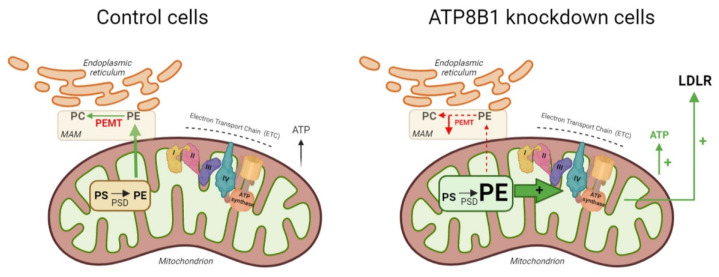
Hypothetical model summarizing the findings described. In mitochondria of control cells (**left**), phosphatidylserine (PS) is converted to phosphatidylethanolamine (PE) by phosphatidylserine decarboxylase (PSD). PE translocates to the mitochondrial-associated membrane (MAM), where phosphatidylethanolamine methyltransferase (PEMT) catalyzes the conversion of PE to PC. In ATP8B1 knockdown cells (**right**), PEMT is down-regulated, which leads to elevated mitochondrial PE levels that activate the protein complexes I–IV (IV = ATP synthase) of the electron transport chain (ETC) resulting in elevated OXPHOS and consequent increased ATP production and LDLR expression. Figure was created in BioRender.com.

## Data Availability

The raw datasets generated and/or analyzed during the current study are available from the corresponding author on reasonable request.

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
