# Peer review of "ATP8B1 Deficiency Results in Elevated Mitochondrial Phosphatidylethanolamine Levels and Increased Mitochondrial Oxidative Phosphorylation in Human Hepatoma Cells"

_ijms, 2022, doi:10.3390/ijms232012344_

Round 1

Reviewer 1 Report

By knocking down the gene coding for the phospholipid flippase ATP8B1 in HepG2 cells, Gomez-Mellado et al demonstrated that when the expression of this flippase is diminished, the OXPHOS activity as well as the LDL-receptor expression are significantly increased. This seems be related with mitochondrial fragmentation and mitochondrial phosphatidyl ethanolamine (PE) accumulation probably due to diminished expression of phosphatidylethanolamine N-methyltransferase, which converts mitochondrial PE in phosphatidyl choline (PC)

Even if this work is well conducted and findings are interesting, some general conclusions are not enough supported by results that weaken the overall work.

However, results are interesting and should be useful for the scientific community and this manuscript could be published if authors are able to answer some concerns below.

Major Concerns

1) Authors focused on mitochondrial lipids and found that there were not significant changes when ATP8B1 was knocked down. However, as mutations in the ATP8B1 gene are related with impaired bile formation (ref 5 et 6 of the manuscript), and bile acid synthesis depends on cholesterol import into the mitochondria, it would be crucial to study whether cholesterol levels in mitochondria, as well as the expression of proteins involved in cholesterol import (Star, TSPO, …) are affected by ATP8B1 knockdown.

2) What happened with lipids content and distribution into the plasma membrane? ATP8B1 is involved in PS translocation, then in the “eat me” cell signaling. Is PS localization modified by ATP8B1 knockdown? What happened with apoptosis levels and cell viability?

3) Discussion should include a paragraph about the function of ATP8B1 unrelated with flippase activity. Indeed, reference 10 cited by authors describes a flippase independent function of this protein that is important for apical structuration of epithelial cells.

4) A final schema illustrating the authors’ hypothesis will significantly improve the readers comprehension.

Minor Concerns

1) Figures 2 and 3: 1) rotonone should be changed by rotenone; 2) are authors sure that all values in panel B come from panel A results?

2) Quantities should be separated from units (for instance: line 83, 10mM should be 10 mM)

3) Authors found that mitochondria are more fragmented when ATP8B1 is knocked down. It would be interesting to explore a bit more the mitochondrial structure (round vs elongated shape, cristae structure, etc.)

4) Lines 273-274: authors mentioned that CD36 expression was induced, but state (data not shown). Considering the unlimited capacity to include results in supplemental information online, these results should be shown or withdrawn.

5) Have authors any information about the effect of ATP8B1 knockdown in peroxisome activity?

Reviewer 2 Report

ATP8B1 deficiency results in elevated mitochondrial phosphatidylethanolamine levels and increased mitochondrial oxidative phosphorylation in human hepatoma cells.

This article reveals that ATP8B1 deficiency resulted in the increased levels of mitochondrial oxidation and PE which induced mitochondrial fragmentation and OXPHOS enhancement. Subsequently, ATP8B1 deficiency causes elevated LDLR in hepatoma cells, suggesting that ATP8B1 deficiency may contributes to PFIC1. The authors provide smooth evidence and clear pathway to proof the hypothesis.

There are some questions and suggestions:

1.       Figure 1: the LDLR protein was detected from total cell lysate or cell membrane extract? It is suggested to clarify the source of the protein.

2.       Figure 1D: please show the indication of LDLR-“m” and LDLR-“i”.

3.       Figure 2A and 3A: the baseline expression of shControl group in these two figures were different (about 2-fold changes). Are these shControl cells processed under the same protocol?

4.       Figure 4A: the mitochondrial proteins were detected from total cell lysate or mitochondrial layer extract? It is essential to further determine the levels of these proteins of mitochondrial layer extract directly.

5.       This study suggested that ATP8B1 deficiency leads to the increase of LDLR expression through the elevated levels of mitochondrial oxidation and PE which subsequently induced mitochondrial fragmentation and OXPHOS enhancement. However, there is no direct evidence reveal the connection of the up-regulated OXPHOS and LDLR. Please clarify the issue.

Round 2

Reviewer 1 Report

The manuscript was sufficiently improved to be published in IJMS.

However, authors should include in supplementary material their data concerning no changes in cholesterol levels. As an on-line open journal, IJMS gives authors the possibility to include all data necessary to fully understand the manuscript rationale, then in my opinion, it’s not acceptable to include “data not shown” statements.

Author Response

We thank the reviewer for his/her positive response on our rebuttal and we apologize for not having included the 'not shown' data in the revised manuscript as supplementary figure.

In the revised manuscript we now refer to supplementary figure S3, which shows the cholesterol data of the mitochondria. We have revised the text accordingly.